# Prostate-Specific Membrane Antigen (PSMA) Expression Predicts Need for Early Treatment in Prostate Cancer Patients Managed with Active Surveillance

**DOI:** 10.3390/ijms242216022

**Published:** 2023-11-07

**Authors:** Elham Ahmadi, Simon Wang, Mohammad Gouran-Savadkoohi, Georgia Douvi, Naghmeh Isfahanian, Nicole Tsakiridis, Brent E. Faught, Jean-Claude Cutz, Monalisa Sur, Satish Chawla, Gregory R. Pond, Gregory R. Steinberg, Ian Brown, Theodoros Tsakiridis

**Affiliations:** 1Department of Oncology, McMaster University, Hamilton, ON L8S 4L8, Canada; ahmade12@mcmaster.ca (E.A.); wangs274@mcmaster.ca (S.W.); gouransm@mcmaster.ca (M.G.-S.); georgia_dvs@hotmail.com (G.D.); naghmeh.isfahanian@gmail.com (N.I.); gpond@mcmaster.ca (G.R.P.); 2Department of Medicine, McMaster University, Hamilton, ON L8S 4L8, Canada; gstein@mcmaster.ca; 3Center for Metabolism, Obesity and Diabetes Research, McMaster University, Hamilton, ON L8S 4L8, Canada; 4Department of Health Sciences, Brock University, St. Catharines, ON L2S 3A1, Canada; nicole.tsakiridis@mail.utoronto.ca (N.T.); bfaught@brocku.ca (B.E.F.); 5Department of Pathology and Molecular Medicine, McMaster University, Hamilton, ON L8S 4L8, Canada; jcutz@mcmaster.ca (J.-C.C.); surm@hhsc.ca (M.S.); 6Department of Laboratory Medicine, Niagara Health System, St. Catharines, ON L2S 0A9, Canada; satish.chawla@niagarahealth.on.ca; 7Department of Surgery, Niagara Health System, St. Catharines, ON L2S 0A9, Canada; ian.brown@niagarahealth.on.ca

**Keywords:** PSMA, GLUT1, ACLY, prostate cancer, progression, active surveillance

## Abstract

Metabolic dysregulation is an early event in carcinogenesis. Here, we examined the expression of enzymes involved in de novo lipogenesis (ATP-citrate lyase: ACLY), glucose uptake (Glucose Transporter 1: GLUT1), and folate–glutamate metabolism (Prostate-Specific Membrane Antigen: PSMA) as potential biomarkers of risk for early prostate cancer progression. Patients who were managed initially on active surveillance with a Gleason score of 6 or a low-volume Gleason score of 7 (3 + 4) were accrued from a prostate cancer diagnostic assessment program. Patients were asked to donate their baseline diagnostic biopsy tissues and permit access to their clinical data. PSMA, GLUT1, and ACLY expression were examined with immunohistochemistry (IHC) in baseline biopsies, quantitated by Histologic Score for expression in benign and malignant glands, and compared with patient time remaining on active surveillance (time-on-AS). All three markers showed trends for elevated expression in malignant compared to benign glands, which was statistically significant for ACLY. On univariate analysis, increased PSMA and GLUT1 expression in malignant glands was associated with shorter time-on-AS (HR: 5.06, [CI 95%: 1.83–13.94] and HR: 2.44, [CI 95%: 1.10–5.44], respectively). Malignant ACLY and benign gland PSMA and GLUT1 expression showed non-significant trends for such association. On multivariate analysis, overexpression of PSMA in malignant glands was an independent predictor of early PC progression (*p* = 0.006). This work suggests that the expression of metabolic enzymes determined by IHC on baseline diagnostic prostate biopsies may have value as biomarkers of risk for rapid PC progression. PSMA may be an independent predictor of risk for progression and should be investigated further in systematic studies.

## 1. Introduction

Prostate cancer (PC) is the second most common solid tumor and the fifth leading cause of cancer mortality in men worldwide [1]. Currently, many low-risk and early-stage localized PC patients do not receive immediate therapy but are monitored with systematic active surveillance programs until disease progression [2]. Active surveillance trials in the past 20 years have shown this is a safe population-based approach for managing low-risk PC. Such studies demonstrated a 25–45% risk of disease progression for low- and intermediate-risk patients over 5–15 years, respectively [3]. The average risk of dying from PC in men on active surveillance has been reported to be low (0.1–5.7%) over 10–15 years [4]. However, the overall number of patients currently managed with this approach is high and continues to increase rapidly. Therefore, a substantial number of PC patients managed with active surveillance today are at risk for rapid disease progression to locally advanced or metastatic PC, which requires treatment with toxic therapies and may be incurable. Developing tools to predict which men are at risk for rapid PC progression can provide significant benefits to active surveillance programs.

While useful at determining disease risk at diagnosis, the current standard of care PC biomarkers, such as prostate-specific antigen (PSA) and tumor Gleason score, have limited value in predicting future risk of disease progression. PSA is not a cancer-specific biomarker, as other conditions, including prostatic inflammation, can regulate it [5,6]. Gleason score is the most widely accepted tissue biomarker for risk stratification and monitoring of active surveillance patients [4]. However, its ability to predict disease outcomes is also limited, especially when used without considering PSA and clinical T-score [7]. Additional blood and urine biomarkers include the Prostate Health Index (PHI), four-kallikrein (4K) Score, and Prostate Cancer Antigen 3 mRNA, which appear to have some predictive value for identifying active surveillance patients at risk for PC progression [8,9]. Modern gene expression tissue biomarkers, such as Oncotype DX, GPS, and Decipher, are now available and may have value in selecting patients [8,9]. However, genomic tests are commercially available in only a few countries and require complex and expensive technologies for their application.

Given the global prevalence of PC, valuable active surveillance biomarkers should be easily adaptable in the routine clinical practice of most institutions. To date, only a few immunohistochemically assessable markers, such as Ki-67, Bcl-2, p53, PTEN, and ERG, have been suggested to have prognostic value in PC, including in active surveillance cohorts [10,11,12]. Nevertheless, these markers have not been incorporated into clinical practice, and the need for convenient and effective biomarkers in this clinical space remains.

Dysregulation of metabolism is a hallmark of cancer. It is modulated in the early stages of carcinogenesis and responds to tumor progression with elevated glucose uptake rates, de novo lipogenesis, and protein synthesis [13]. During progression, PC increases its reliance on glucose uptake and glycolysis [14]. This typically involves enhanced expression of facilitative glucose transporter proteins (GLUTs) on the cell surface, the first rate-limiting step in glucose utilization. Increased expression of GLUT1, encoded by the *SLC2A1* (solute carrier family 2 member 1) gene, has been associated with poor differentiation, larger tumor size, lymph node and distant sites of metastasis, and poor overall and disease-free survival in several tumor types [15,16]. PC studies showed a relationship between GLUT1 overexpression and clinicopathological features of poor prognosis, including high tumor grade, earlier biochemical recurrence, castration resistance, and metastasis [17,18,19,20].

The rapid growth of cancer cells also requires enhanced de novo fatty acid synthesis to build cellular membranes, support cellular integrity, and propagate signaling events [21,22]. The first substrate of de novo lipogenesis is citrate, which is converted to acetyl-CoA by ATP-citrate lyase (ACLY), encoded by the *ACLY* (ATP citrate (Pro-S) lyase) gene [23]. ACLY overexpression was suggested to be a negative prognostic factor in many tumor types [24,25]. Inhibition of ACLY reduced tumor growth and metastasis in various cancers, including PC [26], and sensitized castration-resistant PC cells to androgen receptor antagonists [27]. Despite evidence of enhanced expression of lipogenic enzymes in PC cells [28,29,30,31], ACLY expression has not been investigated in the early stages of PC progression.

Prostate-specific membrane antigen (PSMA), encoded by the *FOLH1* (folate hydrolase 1) gene, is a type II transmembrane glycoprotein [32] with folate hydrolase activity, which releases glutamate from vitamin B9 and other glutamate substrates [33]. It is expressed in benign prostate epithelial cells, prostatic intraepithelial neoplasia (PIN), malignant prostate glands, and metastatic sites [34]. PSMA expression increases with disease progression from early neoplastic lesions to malignant glands and from localized tumor to lymph node and distant metastasis [35]. The expression of this enzyme is suppressed by androgen receptor signaling, and it is upregulated by androgen deprivation. Consistently, PSMA overexpression is associated with androgen-independent growth, advanced tumor stage, higher Gleason grade, and early recurrence after treatment [36,37,38,39,40,41]. Importantly, PSMA is now the target of several diagnostic and therapeutic probes, including ^68^Ga-PSMA-11- and ^18^F-DCFPyL—positron emission tomography (PET) probes used in PC staging, intravenous ^99^mTc-PSMA used in radio-guided surgery, and Lutetium-177-PSMA radio-ligands used in theranostics [42].

Given the evidence discussed, we pursued a pilot prospective investigation of PSMA, GLUT1, and ACLY expression in baseline diagnostic transrectal ultrasound (TRUS)-guided biopsies as prognostic markers of risk for rapid PC progression in patients managed with active surveillance.

## 2. Results

### 2.1. Patients, Tissues, and Clinicopathological Data

A total of 40 patients were recruited for this study from the PC diagnostic assessment program (PDAP) clinic, Niagara Health System, St. Catharines, Ontario, Canada. After initial evaluation with H&E staining, 34 (85%) of 40 patients had tissue satisfactory for pathologic evaluation and scoring. Each patient had at least 1 and, at most, 6 paraffin blocks containing malignant glands on their baseline biopsies. The average number of blocks with malignant glands was 2.9 blocks per patient.

At the time of initial diagnosis, the mean age of patients was 71.3 years, and the mean serum PSA level was 6.43 ng/mL. Of the 34 evaluable cases, 24 (71%) had a Gleason score of 3 + 3 = 6 (ISUP grade group 1) diagnosis, and 10 (29%) had a Gleason score of 3 + 4 = 7 (ISUP grade group 2) (Table 1). Only five patients did not receive definitive therapy by the timepoint data were censored. All patients determined to have disease progression were treated with definitive radiotherapy.

### 2.2. Patterns of Marker Expression

In most patients, PSMA showed a heterogeneous expression pattern in prostatic glands, with apical accentuation in the strongly expressed glands, while GLUT1 and ACLY expression in glands was uniform (Figure 1). PSMA and GLUT1 expression was membranous and cytoplasmic, while ACLY showed only cytoplasmic distribution. PSMA expression (with different intensities) was only observed in prostatic epithelial cells/glands. On the other hand, GLUT1 and ACLY expression were noted in both epithelial and stromal elements. In prostatic epithelial cells, GLUT1 and ACLY had dominant expression with mild and scattered expression in fibromuscular stromal tissue, except in the nerve bundles, which demonstrated strong expression (Appendix A).

### 2.3. IHC Evaluation

Overall, within each patient, PSMA, GLUT1, and ACLY expressions were higher in malignant compared to benign epithelial cells, with a statistically significant difference for ACLY (*p* = 0.04) (Figure 2).

We observed overexpression (H-score of ≥200) of PSMA, GLUT1, and ACLY in malignant glands of 6 (18%), 13 (38%), and 15 (44%) patients, respectively. Figure 3a–c illustrates representative cases of low-expression versus over-expression for each marker in malignant glands. Appendix A compares mean ± SD values of clinicopathologic features (PSA, Gleason score, and time-to-treatment) in each marker’s two expression categories (overexpression: No vs. Yes). Although there were trends for high PSA values and shorter time remaining on active surveillance (time-on-AS) in patients demonstrating marker overexpression, we only observed a statistically significant increase in average serum PSA with patients demonstrating ACLY overexpression (*p* = 0.02, Appendix A).

Univariate analysis showed that PSMA and GLUT1 expression in malignant glands was associated with time-on-AS. Patients overexpressing PSMA or GLUT1 in malignant glands had significantly shorter time-on-AS compared to those with low expression (*p* = 0.002 and 0.03, respectively). However, this could not be shown for ACLY (*p* = 0.38) (Table 2). Figure 3 shows K–M curves illustrating the dichotomy in the rates of progression over time in patients demonstrating low expression vs. overexpression of each marker.

We also examined the relationship between time-on-AS and marker expression in benign glands. Marker expression in benign glands showed trends for association with time-on-AS but did not reach statistical significance (*p* = 0.18, 0.29, 0.32, for PSMA, GLUT1, and ACLY, respectively) (Appendix A and Table 2).

Importantly, multivariate analysis demonstrated that PSMA overexpression in malignant glands (H-score ≥ 200) was an independent predictor of time-on-AS. Patients with PSMA overexpression had a significantly shorter time-on-AS compared to those with low PSMA expression (HR: 4.22, (CI 95%:1.52,11.67), *p* = 0.006) (Table 2).

## 3. Discussion

The most significant finding of the present study is the novel observation that PSMA overexpression in malignant glands of baseline prostate biopsy tissue is an independent predictor of shorter time to progression in active surveillance patients. These observations are consistent with the proposed metabolic function of PSMA in PC. Increased expression and enzymatic activity of PSMA can contribute to a selective growth advantage supporting rapid disease progression [43]. Our findings may have significant clinical implications given the established role of PSMA imaging in advanced prostate cancer. Since PSMA-PET identifies both metastatic and intra-prostatic disease, future studies may consider investigating the additive value of combined PSMA IHC with PSMA-PET imaging to risk-stratify and monitor disease progression in active surveillance patients.

Consistent with reports associating GLUT1 expression with poor PC prognosis and aggressive PC behavior [17,20,44], we found in our cohort that high expression of GLUT1 in malignant glands significantly correlated with shorter time to progression in active surveillance patients. This was detected in univariate but not in multivariate analysis. Immunohistochemical analysis of GLUT1 should be investigated in additional active surveillance cohorts. Further, GLUT1 IHC could be examined in combination with ^18^F-2-deoxy-D-glucose (FDG)-PET imaging to identify patients at risk for rapid disease progression.

We found only statistically insignificant trends for an association between increased PSMA or GLUT1 expression in benign epithelial glands with a risk of short time-on-AS. Biomarker expression in benign glands has not been investigated in active surveillance. Nevertheless, such a finding would be consistent with the notion that benign glands are at risk of transformation into malignant and invasive tumors [45]. Therefore, benign gland PSMA and GLUT1 expression patterns warrant further investigation.

Although earlier studies suggested that ACLY activity supports PC progression and that its expression is associated with a higher Gleason score [46], we could not detect a significant association of ACLY expression with time-on-AS. We found that patients with higher ACLY expression had higher PSA levels and detected only trends for the association of malignant gland ACLY overexpression with shorter time-on-AS.

### Study Limitations

The findings of this work are consistent with the role of PSMA, GLUT1, and ACLY in PC biology [14,28,29,30,33,47]. Nevertheless, the present work is merely a pilot investigation, and our observations require validation in larger active surveillance cohorts.

We made efforts to reduce bias in this analysis using an independent collection of biomarker and clinical data by two separate studies and teams. Due to the retrospective nature of the study of collecting clinical outcomes, we were able to record accurately the time of treatment delivery in patients that progressed and required treatment, but we could not accurately estimate the exact time disease progression was declared by treating clinicians. Nevertheless, wait times for radiotherapy delivery in our institution never exceeded eight weeks. Therefore, given that the present study monitored active surveillance patients over a period of 10 years and treatment events spread over a six-year period, it is highly unlikely that selecting time-on-AS as the primary variable introduced significant bias in our analysis.

## 4. Methods and Materials 

### 4.1. Patient Population and Tissue Collection

With appropriate ethics approval (Hamilton integrated Research Ethics Board (HiREB)-18-039), active surveillance patients were accrued prospectively from the Niagara Health System, Ontario, Canada, PC diagnostic assessment program (PDAP) clinic. All patients had a multidisciplinary clinic visit with both urology and radiation oncology prior to making their treatment choice. Patient informed consent was obtained to have access to clinical information and to provide their formalin-fixed, paraffin-embedded blocks of prostate biopsies for analysis. All patients were approached for accrual at the time of their return to PDAP for standard re-biopsy as part of the active surveillance program.

At the time of accrual, all patients had a baseline diagnosis of Gleason score 3 + 3 = 6 (ISUP grade group 1) or score 3 + 4 = 7 (ISUP grade group 2) adenocarcinoma, acinar type, and serum PSA level < 20 ng/mL (Table 1). The primary outcome in this study was time to disease progression, leading to treatment utilization reported as time-on active surveillance (AS) (time-on-AS). The treating clinicians determined disease progression and treatment utilization at subsequent clinical follow-up visits and were independent of this biomarker analysis study. Events of disease progression determined by treatment utilization were collected independently by an independent study that reviewed NHS-PDAP active surveillance long-term outcomes (HiREB-13-312). The biomarker analysis and active surveillance outcomes study personnel and data collection processes were blinded to each other. Progression events were defined as the detection of:(i)A higher Gleason score in subsequent biopsies, increasing from Gleason 6 to 7 or from 7 to 8(ii)Substantial biochemical progression of PSA to >15 ng/mL(iii)Combined events of >50% increase in the number of involved biopsy cores and PSA progression to >10 ng/mL, or any clinical or imaging indications of progression after initial diagnosis

Analysis of biomarker expression was carried out only on the baseline diagnostic biopsies patients underwent before study accrual.

### 4.2. H&E and Immunohistochemistry (IHC) Staining

Formalin-fixed and paraffin-embedded blocks of prostate core biopsies were serially sectioned into 5 μm thickness. The first section was H&E stained (Abcam#245880, Abcam, Cambridge, UK) to evaluate whether the remaining tissue was satisfactory for following IHC staining analysis. If a patient had multiple malignant blocks, the 3 blocks with higher tumor grade and higher tumor percentage/involvement were selected for IHC staining and scoring.

Tissues were deparaffinized and rehydrated in MQ400 xylene (Sigma-Aldrich#1.08298.4007, Sigma-Alrich, St. Louis, MO, USA) and 95% ethanol (Fisher Scientific#HC-1100-1GL, Fisher-Scientific, Hampton, NH, USA), followed by endogenous peroxidase (Fisher Scientific#H325-500, Fisher-Scientific, Hampton, NH, USA) removal and heat antigen retrieval in citrate buffer with pH:6 (Sigma-Aldrich#C9999, Sigma-Alrich, St. Louis, MO, USA). Then, they were blocked in 10% normal goat serum (Vector Laboratories#S-1000-20, Vector Laboratories, Newark, CA, USA) for 2 hours and incubated overnight at 4 °C with either non-specific (negative control) serum or rabbit monoclonal PSMA antibody (Cell Signaling#12815, Cell Signaling, Danvers, MA, USA) with 1/100 dilution, GLUT1 antibody (Abcam#115730, Abcam, Cambridge, UK) with 1/500 dilution, and ACLY antibody (Abcam#40793, Abcam, Cambridge, UK) with 1/200 dilution, followed by 1:500 biotinylated goat-anti-rabbit IgG secondary antibody (Vector Laboratories#BA-1000, Vector Laboratories, Newark, CA, USA) and 1:50 streptavidin peroxidase (Vector laboratories#SA-5004, Vector Laboratories, Newark, CA, USA), and developed using Nova Red kit (Vector laboratories#SK-4800, Vector Laboratories, Newark, CA, USA). Hematoxylin (Abcam#245880, Abcam, Cambridge, UK) was used as a counterstain.

#### Special Consideration for the Antibody Staining Procedure

Before IHC staining on prostate core needle biopsy, the technique and all the reagents were first validated on radical prostatectomy tissue samples to ensure the technique and all the reagents were working properly. Also, the dilutions of primary antibodies were optimized on radical prostatectomy tissue samples first. Negative and positive controls (internal and/or external) were included in every round of IHC staining for quality assurance.

### 4.3. Scoring

The widely used H-Score system was employed for IHC evaluation [48,49]. This method of quantification is determined by multiplying the percentage of cells with cytoplasmic staining by their intensity in ordinal values (0 = no, 1 = weak, 2 = medium, 3 = strong staining) (Appendix A) [(3 × % of strongly staining nuclei) + (2 × % of moderately staining nuclei) + (1 × % of weekly staining nuclei)], which ranges from 0 to 300 possible values [50]. We determined the cut-off value for marker overexpression versus non-overexpression to be an H-score of 200 contrasted with <200.

Benign glands/epithelium and malignant glands were identified and scored separately by clinically trained pathologists. Atrophic glands/epithelium and high-grade PIN were excluded from scoring. All slides were evaluated with AMACR (α-Methylacyl-CoA Racemase) IHC staining to confirm the malignant nature of glands [45] (Appendix A). H-Scoring was performed in a blinded fashion using an Olympus BX-40 microscope with no previous knowledge of the patient’s history. If there was more than one block for a patient evaluated for IHC staining and scoring, the highest score per patient was considered.

### 4.4. Statistical Analyses

Independent *t*-tests were used to examine the equality of the means between PSMA, GLUT1, and ACLY overexpressing and non-overexpressing groups. Statistics were performed using GraphPad Prism 8 (San Diego, CA, USA) with a significance level of 0.05. Disease progression-free analysis, indicated by the variable of time on active surveillance (time-on-AS) (the interval between initial biopsy-proven diagnosis of PC and treatment initiation), was estimated using the Kaplan–Meier method. Cox proportional hazards regression was used to evaluate the prognostic ability of factors on time-on-AS. Multivariate analysis and two-sided confidence intervals were constructed for outcomes of interest.

## 5. Conclusions

This work suggests that elevated PSMA expression in malignant glands of baseline diagnostic prostate biopsies may be able to independently predict risk for early PC progression in patients managed with active surveillance. Elevated expression of GLUT1 in malignant glands and marker expression in benign glands warrants further investigation. Immunohistochemical analysis of metabolic markers can be easily incorporated into the clinical practice of most institutions. Therefore, validation of our results within larger prospective studies may lead to improvements in the management of active surveillance patients on a global scale.

## Figures and Tables

**Figure 1 ijms-24-16022-f001:**
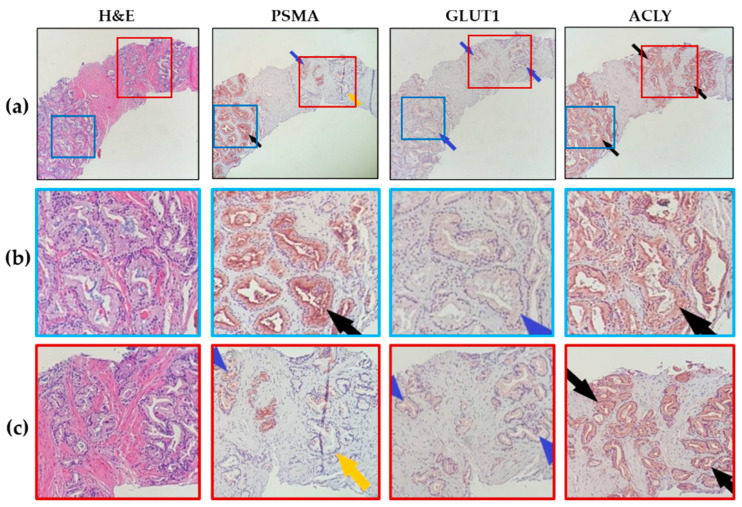
**Patterns of marker expression in prostate biopsies.** (**a**) Representative images (10× magnification) showing heterogeneous PSMA expression but uniform GLUT1 and ACLY expression in prostate glands. Mild expression for GLUT1 and strong expression for ACLY are shown in this representative case. (Black arrows point to moderate to strong marker expression, blue arrows for mild marker expression, and yellow arrows point to no expression). (**b**,**c**) Magnified views of images in (**a**).

**Figure 2 ijms-24-16022-f002:**
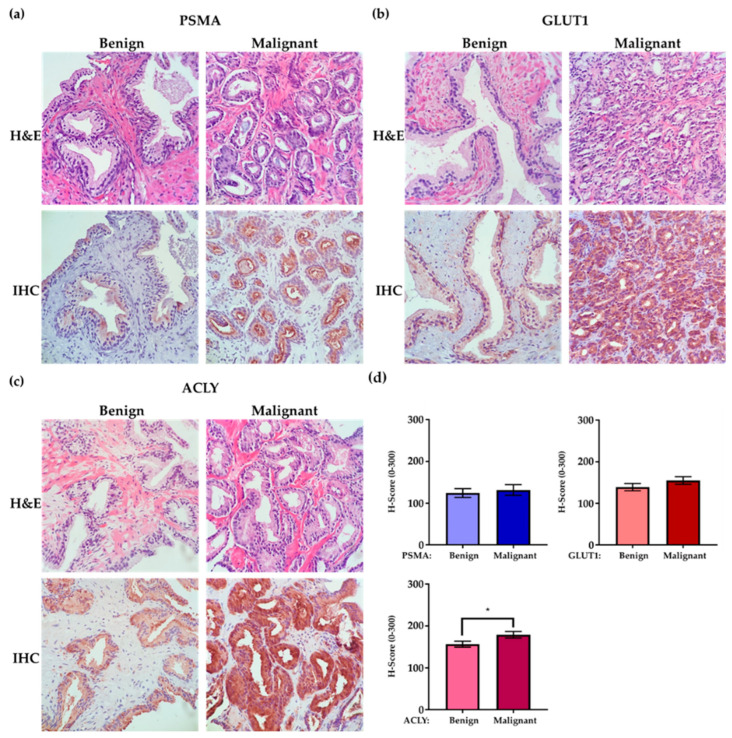
**Enhanced PSMA, GLUT1, and ACLY expression in malignant prostate glands compared to benign glands.** Prostate biopsy tissue, including benign and malignant components (40× magnification). (**a**) Prostate biopsy tissue of a 71-year-old patient, GS = 3 + 3 = 6 (ISUP Grade Groupe 1), reveals higher PSMA expression in prostate adenocarcinoma compared to benign epithelial cells. (**b**) Prostate biopsy tissue of a 73-year-old patient, GS = 3 + 4 = 7 (ISUP Grade Groupe 2), shows enhanced GLUT1 expression in prostate adenocarcinoma compared to benign epithelial cells. (**c**) Prostate biopsy tissue of a 76-year-old patient, GS = 3 + 3 = 6 (ISUP Grade Groupe 1), reveals enhanced ACLY expression in malignant glands compared to benign glands. (**d**) Graphs illustrate average H-scores of PSMA, GLUT1, and ACLY IHC expression in benign and malignant glands for the entire group of patients (error bars are SEM, * = *p*-value 0.04).

**Figure 3 ijms-24-16022-f003:**
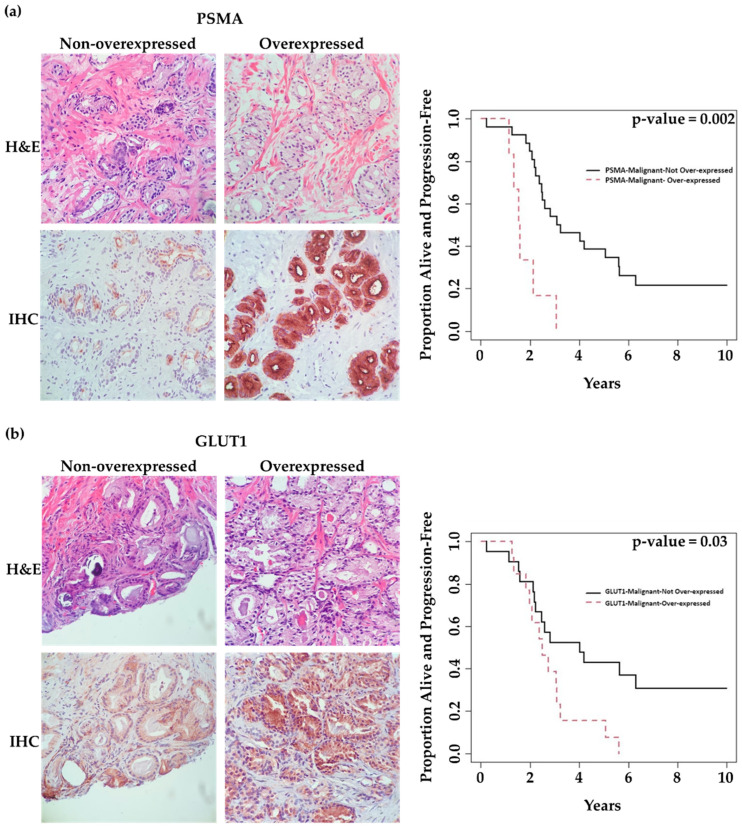
Association of marker expression in malignant prostatic glands with time patients remained on active surveillance (time-on-AS). (40× magnification) (**a**) Images on the left: 64-year-old patient, GS = 3 + 4 = 7 (ISUP grade group 2), revealing low tumoral PSMA expression with a long time-to-progression (124.1 months). Images on the right: 75-year-old patient, GS = 3 + 4 = 7 (ISUP grade group 2), revealing tumoral PSMA overexpression with a short time to progression (16.1 months). (**b**) Images on the Left: 64-year-old patient, GS = 3 + 4 = 7 (ISUP grade group 2), revealing low tumoral GLUT1 expression with a long time to progression (124.1 months). Images on the Right: 75-year-old patient, GS = 3 + 4 = 7 (ISUP grade group 2), revealing tumoral GLUT1 overexpression with a short time to progression (22.2 months). (**c**) Images on the Left: 84-year-old patient, GS = 3 + 3 = 6 (ISUP grade group 1), revealing low tumoral ACLY expression with a long time to progression (75.6 months). Images on the Right: 76-year-old patient, GS = 3 + 3 = 6 (ISUP grade group 1), revealing tumoral ACLY overexpression with a very short time to progression (2.8 months). Kaplan–Meier survival curves in relation to marker expression in the malignant components of prostate biopsies.

**Table 1 ijms-24-16022-t001:** Patient clinical characteristics.

Clinical Characteristics	Mean (Range)	
Age at diagnosis (years)	71.3 (52–84)	
Serum PSA level (ng/mL)	6.43 (1.80–15)	
Prostate Cancer Grade	**ISUP Grade Group (Gleason Score)**	**N (%)**
	1 (3 + 3 = 6)	24 (71%)
	2 (3 + 4 = 7)	10 (29%)

**Table 2 ijms-24-16022-t002:** Association between time to progression and PSMA, GLUT1 and ACLY expression status (<200 vs. ≥200).

Variable	HR (95% CI)	*p*-Value
**Univariate analysis**		
PSMA-Malignant	5.06 (1.83, 13.94)	0.002
PSMA-Benign	2.10 (0.71, 6.26)	0.18
GLUT1-Malignant	2.44 (1.10, 5.44)	0.03
GLUT1-Benign	1.65 (0.65, 4.16)	0.29
ACLY-Malignant	1.40 (0.66, 2.95)	0.38
ACLY-Benign	0.65 (0.27, 1.53)	0.32
**Multivariate analysis**		
PSMA-Malignant	4.22 (1.52, 11.67)	0.006
GLUT1-Malignant	1.95 (0.84, 4.51)	0.12

## Data Availability

Data is contained within the article or Appendix A. Requests for additional data may be sent to the corresponding author.

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
