# Peer review of "Prostate-Specific Membrane Antigen (PSMA) Expression Predicts Need for Early Treatment in Prostate Cancer Patients Managed with Active Surveillance"

_ijms, 2023, doi:10.3390/ijms242216022_

Round 1

Reviewer 1 Report

Comments and Suggestions for Authors

1.     Please put the gene names in italics throughout the manuscript.

2.     In line 145, please rephrase the sentence keeping the patient numbers either numeric or alphabetic.

3.     In line 192, ‘shorted’ should be rephrased as ‘shorter’.

4.     In the ‘Discussion’ section, please provide a subsection with the heading ‘Limitations of the study’ and discuss it appropriately.

5.     The manuscript should have a ‘Conclusion’ subheading, which will attract the reader’s attention.

6.     In the ‘Materials and Method’ section and specifically in ‘4.2. H&E and Immunohistochemistry (IHC) staining’, please provide the RRID against every reagent, and antibody used. Also, mention if there was any special consideration for the antibody staining procedure.

Author Response

We thank the reviewer for the careful review of our manuscript. Below we provide our response to reviews and list the modification made to the manuscript.

Reviewer 1

  1. Please put the gene names in italics throughout the manuscript.

This correction has been made

  1. In line 145, please rephrase the sentence keeping the patient numbers either numeric or alphabetic.

This issue has been resolved. As per comments of other reviewers, the results sections has been modified extensively to improve falrity.

  1. In line 192, ‘shorted’ should be rephrased as ‘shorter’.

The error has been corrected.

  1. In the ‘Discussion’ section, please provide a subsection with the heading ‘Limitations of the study’ and discuss it appropriately.

Thank you for this point: A new heading is inserted and the discussion on limitations is expanded

  1. The manuscript should have a ‘Conclusion’ subheading, which will attract the reader’s attention.

Thank you for this point: a new heading is inserted

  1. In the ‘Materials and Method’ section and specifically in ‘4.2. H&E and Immunohistochemistry (IHC) staining’, please provide the RRID against every reagent, and antibody used. Also, mention if there was any special consideration for the antibody staining procedure.

This point has been addressed, the required information has been inserted.

Reviewer 2 Report

Comments and Suggestions for Authors   This study is relevant and of general interest to the readers of this journal.
The title reflects the article content and helps the readers navigate through the article.
The abstract contains all the necessary information in a concise form.
The "Introduction" section is clear and easy to read.
The "Materials and Methods" section is well-described.
The "Results" section could benefit from a bit more detail.
There is no title, and a brief legend for the attached figures is missing.
In the "Discussion" section, it would be interesting to mention the practical applicability of this study in clinical practice and if there are any limitations and potential biases in the conducted study.

Kind regards

Comments on the Quality of English Language

Minor editing of English language required

Author Response

Reviewer 2

We thank the reviewer for the careful review of our manuscript.

In response to the comments of all reviewers, we have now edited the manuscript extensively to improve clarity and to address comments.

Below we provide our response to reviews and list the modifications made to the manuscript.

The new edits-tracked version of the manuscript illustrates all modification made.

____________________

Comments and Suggestions for Authors

This study is relevant and of general interest to the readers of this journal.
The title reflects the article content and helps the readers navigate through the article.
The abstract contains all the necessary information in a concise form.
The "Introduction" section is clear and easy to read.
The "Materials and Methods" section is well-described.

The "Results" section could benefit from a bit more detail.

We appreciate the positive comments of the reviewer. In response to the reviewer’s comments, we have edited extensively the results section to improve clarity. A new figure (Figure 1) is now inserted to illustrate the results discussing the general pattern of marker distribution within biopsy tissues.

There is no title, and a brief legend for the attached figures is missing.

Figure legends have been edited to improve clarity.

In the "Discussion" section, it would be interesting to mention the practical applicability of this study in clinical practice and if there are any limitations and potential biases in the conducted study.

The discussion section has been edited extensively to highlight the limitations of this study and the potential of this work to influence clinical practice if the results of this pilot study could be validated in larger cohorts of active surveillance patients.

Language has been edited throughout the manuscript to improve clarity.

Reviewer 3 Report

Comments and Suggestions for Authors

Introduction: The authors should improve the introduction, making clearer what is the gap in the literature that is filled with this study.

Tables need to reformate

some supplementary must inert in the original paper

At the time of initial diagnosis, the mean age of patients was 111 71.3 years, and the mean serum PSA level was 6.43ng/ml.  must transfer to methods

why select all  patients at 71.3 years,

Comments on the Quality of English Language

 Minor editing of English language required

Author Response

Reviewer 3

We thank the reviewer for the careful review of our manuscript. In response to the comments of all reviewers, we have now edited the manuscript extensively to improve clarity and to address comments.

Below we provide our response to reviews and list the modifications made to the manuscript.

The new edits-tracked version of the manuscript illustrates all modifications made.

Comments and Suggestions for Authors

Introduction: The authors should improve the introduction, making clearer what is the gap in the literature that is filled with this study.

The introduction section and the rest of the manuscript has been edited extensively to improve clarity and highlight the large number of patients this type of work can affect, the lack of useful IHC-based biomarkers in active surveillance, and the need to develop such markers.

Tables need to reformate:

Table 1 has been edited to improve clarity.

some supplementary must inert in the original paper

We have now inserted elements of supplemental data into the regular manuscript. A new Figure 1 is inserted illustrating points discussed in the first part of the “Results” section.

At the time of initial diagnosis, the mean age of patients was 111 71.3 years, and the mean serum PSA level was 6.43ng/ml.  must transfer to methods

why select all  patients at 71.3 years,

We are not clear on the point raised here by the reviewer. As indicated above, Table 1 is now edited to improve clarity.

Language has been edited throughout the manuscript to improve clarity.

Round 2

Reviewer 3 Report

Comments and Suggestions for Authors

Thank you very much for correcting the research article 

Please, find my comments and suggestions below.

1- line 45, authors start with word currently while reference 2018 Komisarenko M, Martin LJ, Finelli A. Active surveillance review: contemporary selection criteria, follow-up, compliance and 383 outcomes. Transl Androl Urol. Apr 2018;7(2):243-255. doi:10.21037/tau.2018.03.02

please cite reference 2023

2- Table 1 need need needs to be reformate, there are rows nil, so write the head titles of rows 

3-  kindly recommend using the MDPI styles and formatting for each element of the paper: text (with paragraph indents), headings, figure captions, reference style, etc. Please, use the formats included into the MDPI template.

4- Please, add the grade (quality) of the chemicals used.

5-  Moreover, in the  current form the Conclusion is very general. Please, rewrite using the quantitative parameters. Moreover, the planned investigations could be added in this section.

6- Avoid overly long sentences and paragraphs. Break them down into shorter, more focused sentences for improved clarity

Comments on the Quality of English Language

Extensive editing of English language required

Author Response

We thank the reviewer for the careful and thoughtful comments on the last version of our manuscript. We have reviewed, edited and reformatted the manuscript in response to those comments. We hope this version of the manuscript is acceptable for publication.

 Response to comments

1- line 45, authors start with word currently while reference 2018 Komisarenko M, Martin LJ, Finelli A. Active surveillance review: contemporary selection criteria, follow-up, compliance and 383 outcomes. Transl Androl Urol. Apr 2018;7(2):243-255. doi:10.21037/tau.2018.03.02

please cite reference 2023

This reference has been updated and all references were adjusted and were reformatted.

 2- Table 1 need need needs to be reformate, there are rows nil, so write the head titles of rows 

3-  kindly recommend using the MDPI styles and formatting for each element of the paper: text (with paragraph indents), headings, figure captions, reference style, etc. Please, use the formats included into the MDPI template.

 All tables and figures, headings figure captions and references were reformatted to follow the MDPI template.

4- Please, add the grade (quality) of the chemicals used.

We have edited the methods section on this point to the extend information is available.

5-  Moreover, in the  current form the Conclusion is very general. Please, rewrite using the quantitative parameters. Moreover, the planned investigations could be added in this section.

We have given this point a thorough consideration. However, we feel that we should be careful to not over-interpret the results of our study. Although our findings are very interesting, we feel that it is important to stress the fact that, due to the small sample size, this is merely a pilot analysis that should trigger further investigation of PSMA and other metabolic markers in larger active surveillance cohorts. As such, we think that stating quantitative results in the conclusions section is not critical for this study, but it is rather important to stress the fact that our results are validated in larger cohorts.

6- Avoid overly long sentences and paragraphs. Break them down into shorter, more focused sentences for improved clarity.

We thank the reviewer for this comment. We have made additional edits throughout the manuscript to reduce sentence length and improve clarity.

Round 3

Reviewer 3 Report

Comments and Suggestions for Authors

thanks for the authors  most correctiobn s were done but  minor corrections as follow  

1-Table 1. Clinical characteristics of ..........  complete 

2- Table 2)  must change to Table 2.

3- line 298 (& ) change to and 

Comments on the Quality of English Language

 Minor editing of English language required

Author Response

thanks for the authors  most correctiobn s were done but  minor corrections as follow  

Thank you for the additional comments.

We have address those comments as indicated below:

1-Table 1. Clinical characteristics of ..........  complete 

We have edits this Table Legend to :  “Patient Clinical characteristics”

2- Table 2)  must change to Table 2.

This has been corrected

3- line 298 (& ) change to and 

We are a little unclear on this point.  The term “H&E” is a standard terms used wide pathology,  we are unclear as to whether the reviewer ask us to replace it with “HandE”, which we have not encountered before. No change has been made to this section.

Minor editing of English language required

We have further edited the language particularly in the methods section to improve clarity.